# Prognostic Factors for Functional Outcome of Patients with Optic Nerve Sheath Meningiomas Treated with Stereotactic Radiotherapy–Evaluation of Own and Meta-Analysis of Published Data

**DOI:** 10.3390/cancers13030522

**Published:** 2021-01-29

**Authors:** Bogdan Pintea, Azize Boström, Sotiris Katsigiannis, Konstantinos Gousias, Rares Pintea, Brigitta Baumert, Jan Boström

**Affiliations:** 1Department of Neurosurgery, BG Universitätsklinikum Bergmannsheil Bochum, Bürkle-de-la-Camp-Platz 1, 44789 Bochum, Germany; 2Department of Radiosurgery and Stereotactic Radiotherapy, MediClin Robert Janker Clinic and MediClin MVZ Bonn, Villenstrasse 8, 53129 Bonn, Germany; Azize.Bostroem@mediclin.de (A.B.); brigitta.baumert@ksgr.ch (B.B.); jan.bostroem@ruhr-uni-bochum.de (J.B.); 3Department of Neurosurgery, Universitätsklinkum Knappschaftkrankenhaus Bochum, In der Schornau 23-25, D-44892 Bochum, Germany; sotiriskatsigiannis@yahoo.gr; 4Department of Neurosurgery, St. Marien Hospital Lünen, Altstadtstraße 23, 44534 Lünen, Germany; kostasgousias@yahoo.com; 5Ophtalmological Office Hamm, Luisenstr. 1, 59065 Hamm, Germany; Rares.pintea@yaho.de; 6Department of Radiooncology, Kantonsspital Graubünden, Loestrasse 170, 7000 Chur, Switzerland; 7Department of Radiotherapy and Radio-Oncology, Universitätsklinikum Marien Hospital Herne, Gamma Knife Zentrum Bochum, In der Schornau 23-25, 44892 Bochum, Germany

**Keywords:** optic nerve sheath meningiomas (ONSM), fractionated stereotactic radiotherapy (fSRT), visual acuity

## Abstract

**Simple Summary:**

This article highlights the fractionated stereotactic radiotherapy in the treatment of optic nerve sheath meningioma (ONSM). ONSM, a slow-growing tumor next to the optic nerve, causes over a long period partial or complete blindness. As visual impairments appear and progress slowly in the disease history, the need for treatment and treatment starting time are controversial. We restate in our article that fractionated stereotactic radiotherapy improves visual function. We further search for prognostic factors which might provide valuable additional and independent information on when to start treatment. We reveal that the visual acuity is a prognostic factor for the outcome of stereotactic therapy for ONSM and promote it as a biomarker for the decision for treatment initiation.

**Abstract:**

Objective: To evaluate prognostic factors for a favorable outcome (improvement of the visual acuity or visual fields) after fractionated stereotactic radiotherapy (fSRT) of optic nerve sheath meningioma (ONSM). Methods: We performed a database search for ONSM treatments during the period from April 2008 to September 2019 in the prospective database for stereotactic radiosurgery/radiotherapy (SRS/SRT) of the Robert Janker Clinic Bonn (Department of Radiotherapy) and performed a literature review and meta-analysis of published data on ONSM between 2010 and 2019. Ophthalmic status before and after treatment was evaluated and the collective was dichotomized into two groups: functional improvement (FI; improvement of either visual acuity or visual fields) and non functional improvement (NFI; with stable or deteriorating visual acuity or visual fields). The two groups were compared regarding different variables: pretreatment visual acuity, age, gender, gross tumor volume (GTV), follow up (FU) time, tumor localization, and maximal retina dose. Results: Overall, 13 stereotactic radiotherapies were performed for ONSM (12 × fSRT, 1 × SRS). Mean follow up was 3 years (range: 1–5 years). The total dose was 50.4 Gy (5 × 1.8 Gy/week) in 12 patients treated with fSRT and 1 × 14 Gy in one SRS case. Mean GTV was 1.13 ccm (range: 0.44–2.20 ccm). During follow up, all tumors were stable or showed shrinkage of tumor volume (100% tumor control), no adverse events were observed, 53% of the patients achieved either better visual acuity or visual fields. Pretreatment visual acuity was significantly different between the FI and the NFI group (0.17 vs. 0.63, *p* = 0.03) in our series and in the meta analysis (*p* < 0.01). Moreover, shorter FU time and lower retinal dose were significantly linked (*p* < 0.05 and *p* < 0.01, respectively) with a better outcome in the meta-analysis but not in our patient cohort. Intracranial tumor localization, gender, and age were not significantly different between the two outcome groups. Conclusion: FSRT for ONSM achieves in over 50% of cases an improvement of the ophthalmic status with low morbidity and excellent tumor control in our series and the meta analysis. Patients with a favorable outcome had in all analysis a significantly higher visual acuity before treatment start. Therefore, we advocate using fSRT as early as possible before vision deterioration occurs.

## 1. Introduction

Optic nerve sheath meningiomas (ONSM), although rare, account for one-third of primary optic nerve tumors [1]. Their circumferential pattern of growth in relation to the optic nerve and the central retinal artery as well as their involvement in the extradural blood supply of the nerve make surgical resection extremely difficult and if attempted it is often associated with increased morbidity and almost no chance for functional improvement either of the visual acuity nor of the visual fields [2,3,4,5,6]. This fact, along with their slow growth rate, has led to the widespread opinion that observation may be the best management. It has been shown though that without any treatment, over years visual impairment occurs in the vast majority of cases accompanied with a substantial risk of involvement of both eyes [2,4,6,7], though ONSM cause no deaths. Resection is preserved for cases with advanced impairment of vision, disfiguring proptosis or those threatening intracranial extension [3].

The need for an effective early therapy option in ONSM to control local tumor growth that preserves and potentially even improves functionality has led to the idea for using fractionated stereotactic radiotherapy (fSRT) [2,3,8]. FSRT is the mindful combination of stereotactic precision with conventional fractionation, combining the benefits of both surrounding tissue protection by the high stereotactical precision and protection of the optic nerve itself by acceptable daily low dose fractions of 1.8 Gy.

FSRT seems to represent a less aggressive alternative to surgery with minimal morbidity and even the chance of improvement of the visual impairment [4,8,9,10,11,12,13]. However, the timing of treatment is still debatable as radiotherapy might have long-term risks of radiogenic cataract, retinopathy and optic neuropathy. Therefore, a watchful waiting strategy is still favored primarily by many physicians with no clear cut agreement as to which stage of visual deterioration radiotherapy would be reasonable with the best risk–benefit ratio. 

In the current study we evaluated factors influencing the functional outcome of ONSM patients treated with fSRT to determine critical clinical values for treatment initiation. 

## 2. Methods

Eligible patients were identified by scanning the database for stereotactic radiotherapy/radiosurgery (SRT/SRS) at the Department for Radiotherapy, MediClin Robert Janker Clinic, Bonn for ONSM treatments. The database contains a total of 273 meningioma cases treated with SRT between April 2008 to September 2019.

### 2.1. Inclusion Criteria

Patients with meningioma originated from the optic nerve sheaths. The ONSM was identified on magnetic resonance imaging (MRI) as hyperintense structure next to the optical nerve on contrast-enhanced T1-weighted images. No biopsy was required for a typical image finding.

### 2.2. Exclusion Criteria

Meningioma patients with secondary involvement of the optic nerve, such as sphenoid wing meningiomas. 

The visual acuity was regularly analyzed before and after SRT/SRS and was prospectively documented yearly. Visual fields were qualitatively evaluated and we distinguished three categories: worse, stable, better. Adverse events were retrospectively evaluated as documented in the patient records or reported post hoc by the practitioners using the Common Terminology Criteria (CTC) for adverse events version 3.0 [14].

### 2.3. Treatment Technique

For treatment preparation a high-resolution 3D gadolinium-enhanced MRI scan of the brain with 1 mm slice thickness were obtained. Immobilization was obtained by using a thermoplastic mask (Brainlab AG, Munich, Germany). Repositioning accuracy was measured by using a KV X-ray image guidance system (ExacTrac, BrainLab AG, Munich, Germany). For treatment planning, a high-resolution CT scan with axial slices of 1.0 mm were obtained and fused with the MRI scan for tumor volume definition using stereotactic treatment planning software (iPlan RT Dose 3.0. BrainLAB AG, Munich, Germany). The gross tumor volume (GTV) was defined as the contrast-enhancing visible optical meningioma. No expansions for a clinical tumor volume (CTV) were made, but an isotropic margin for the planning target volume (PTV) of 1–2 mm was added. Treatment was performed with a dedicated linear accelerator for stereotactic radiotherapy (Novalis classic; BrainLab AG, Munich, Germany) with micro-multileafs (m3 micromultileaf collimator). The treatment technique applied was mainly dynamic arcs combined with static beams intensity-modulated. Dose was prescribed according to ICRU with at least 95% coverage to the tumor volume. 

### 2.4. Outcome Parameters

Local tumor growth on the one hand and functional visual outcome on the other hand were the two main parameters for outcome analysis. The 3D-MRI courses of all ONSM (GTV in ccm) were routinely evaluated volumetrically at every FU. Because of the small collective we dichotomized our collective into patients with functional improvement (FI) (improvement of either visual acuity or of the visual field) and those with no improvement or even deterioration (NFI) of the visual acuity or visual fields. Univariate and multivariate statistical analyses were performed to evaluate patients and tumor-related factors that influenced the functional outcome after SRT/SRS. The following factors were taken into account: patient age, gender, tumor localization: combined intracranial and intraorbital (=posterior) vs. intraorbital (=anterior) site, surgical pretreatment, visual acuity before radiotherapy, FU time and the maximal retinal dose.

Additionally, we reviewed the published data on fractionated SRT of ONSM for the study time period between 2010 and 2019 [5,15,16,17,18,19,20,21,22,23,24,25,26,27]. All data were pooled together and dichotomized as mentioned above. The same univariate and multivariate statistical analyses were performed to evaluate patient- and tumor-related factors that influenced the functional outcome after fSRT of the pooled series.

These statistical comparisons were performed by the Fisher’s exact and Mann–Whitney test analyses. Multivariate analyses were performed by binary logistic regression analysis using the IBM Statistic SPSS 20.0 software. MedCalc v19.0.4 software was used for receiver operating characteristic (ROC) analyses. Significant differences were defined as those with a *p* < 0.05.

## 3. Results

### 3.1. Evaluation of Own Data

Out of 13 patients with ONSM, 12 were treated by fractionated SRT and 1 by SRS (Appendix A). The single case with SRS was a patient with preexisting complete vision lost after microsurgical decompression plus biopsy. Three patients in total received surgical treatment before radiotherapy. After fSRT, tumor control was achieved in all cases, so local tumor control was 100%. We identified 7/13 individuals with functional visual improvement (FI) versus 6/13 cases with no improvement or worsening visual function (NFI).

Mean age was 46 years at start of radiotherapy, with no significant difference between the FI and the NFI group (47 years vs. 46 years). Gender distribution was similar and not significantly different between both groups (57% female in the FI group vs. 40% female in the NFI group) as well as the percentage of the patients with surgical pretreatment (14% in the FI group vs. 30% in the NFI group). Likewise, the maximal retina dose was not significantly different between patients with improvement and those with no improvement or worsening of the ophthalmic status (39.6 Gy in the FI collective vs. 38.9 Gy in the NFI group). We found only one significant difference (*p* = 0.03) between favorable and less favorable cases when the visual acuity before SRT/SRS was taken into account. The mean visual acuity of patients in the FI group was 0.52 and that of patients in the NFI group 0.135. Notably, no patients with a visual acuity less than 0.4 reached a positive outcome. A higher percentage of patients with a combined intraorbital and intracranial tumor site belonged to the group with the less favorable functional outcome (80% NFI group vs. 57% FI group), but this difference did not reach significance. In addition, the functional ophthalmologic outcome seemed to depend on the FU time as improvement appeared after a certain period of 12 to 24 months. This was however not relevant for our evaluation as all of our patients had a FU of more than one year and only the last visual acuity and visual field outcome were taken into consideration (Table 1). 

### 3.2. Review of Published Data and Meta-Analysis

A PubMed literature review from 2010 to 2019 revealed 16 peer-reviewed ONSM publications where 9/16 publications presented individual data for the included patients. All individual data from these earlier publications and our trial were pooled together and prepared for univariate and multivariate statistics. A total of 102 individual data samples were dichotomized regarding visual outcome into two groups: one with favorable functional outcome (FI) (improvement of either visual acuity or of the visual fields) and a second one with no improvement or even deterioration of the visual acuity or visual fields (NFI). A total of 69 patients were in the FI group, while 33 patients were assigned to the NFI group. The variables age, gender, retina dose (Gy), visual acuity before treatment, FU time in years and total radiation dose (Gy) were collected as far as they were documented; further details can be found in the Appendix A.

Univariate analysis (Table 2) revealed for the variable retinal dose a significant change (*p* = 0.02) and for the pretreatment visual acuity a trend for significance (*p* = 0.06). The other variables age, gender, follow up time and total radiation dose were not significantly different between the two groups. A ROC analysis revealed that the optimal threshold might be a pretreatment visual acuity of 0.28, taking in account that higher specificity is desired in our model (Appendix A).

Different multivariate regression analyses (stepwise, enter, backward, forward) regarding visual improvement with the independent variables age, gender, FU time, total radiation dose, retinal dose and pretreatment visual acuity resulted in three significant independent variables: FU time, retinal dose and pretreatment visual acuity (Table 3). A shorter FU time and a lower retinal dose significantly correlated with a better outcome (*p* < 0.05 and *p* < 0.01, respectively), however these correlations had weak partial coefficients (B: -0.15 and B: -0.011, respectively). The pretreatment visual acuity on the contrary had a high partial coefficient of B: +0.60 with a *p* < 0.01, revealing the relevant influence upon the outcome.

## 4. Discussion 

### 4.1. Advantages of fSRT

Our retrospective analysis reflects that treatment of ONSM with fSRT offers a good treatment option with possible improvement of the ophthalmic status in over 50% of the cases, an outcome not reachable by microsurgical treatment [28,29,30]. However, a favorable outcome with some kind of recovery of either the visual acuity or the visual fields might depend on different influencing factors such as age, gender, origin site of the tumor and maximal dose to structures like the retina. However, we could not find any statistical evidence for these hypotheses either in our small series or the meta-analysis of the published data. Some reasons for this failure might be the small collective size of our patient cohort and the unsystematic FU time recording and unsystematic assessment of exact tumor localization in the reviewed data, but perhaps it is also an expression of the lower influence of these factors upon the ophthalmic outcome in contrast to maybe the most important factor: the pretherapeutic impairment of the optic nerve or its clinical correlate, the pretherapeutic visual acuity, as revealed in our series and supported by the review and meta-analysis of the published data. If a certain state of neuropathic damage is reached, a recovery might be more unlikely. This observation is described also for microsurgical management of ONSM. Interestingly, the reported cut off visual acuity of 20/50 for a good outcome is equal to ours [29].

Remarkably, despite some adaptation of stereotactic irradiation parameters to overcome the risk of radiogenic optic neuropathy (=lower total dose concepts=50.4 Gy) the local tumor control of almost 100% for ONSM in our series remains similar to the higher total dose concepts (≥54 Gy) of stereotactic radiotherapy for meningiomas in other localizations. However, extended long-term follow up of more than 10 years has not existed up to now [24]. Nevertheless, taking into account the potential local tumor progress risk after microsurgery of 17.8%, primary stereotactic radiotherapy seems to be the best tumor control option for meningiomas of this localization [26].

### 4.2. Therapeutic Consequences 

In general, the first choice in the treatment of meningioma is microsurgical resection. However, especially at sites nearby or with involvement of critical structures like the optic nerve, the brainstem and in the course of cranial nerves, microsurgical resection has low potential of functional improvement [1,2,3,31]. Taking into consideration the low morbidity profile of fSRT in the treatment of ONSM in our trial as well in other trials, we propagate an early treatment start [4,5,7,8,15,16,17,18,20,21,22,23,24,25,26,32]. However, in very young individuals with a long life expectation it is unknown if fSRT could prevent them from deterioration of their ophthalmic status to the end of their life. Moreover, if treatment is started when patients are oligosymptomatic and visual acuity is almost normal, the balance between treatment benefit and risk might not yet be adequate. In these cases, a watchful waiting strategy might still be reasonable at least if visual acuity is ≥0.8. 

Although our data indicate that a functional improvement can be achieved up to a cut off visual acuity of 0.4, and ROC of the data review analyses reveal a possible threshold at a visual acuity of about 0.28, the authors argue for the earliest possible start of fSRT after individual consideration of the benefits and risks.

A shorter FU time significantly correlates with a better outcome in the multivariate analysis. The correlation is weak, however it should be carefully monitored in following studies as it might reflect a possible radiogenic optic neuropathy or a local tumor progress. 

## 5. Conclusions

Our results reflect that at least for ONSM, fSRT is the treatment option of choice, which can preserve or even improve optic nerve function combined with very good tumor control. Both are treatment goals, which microsurgery has failed to achieve in a similar balanced manner. Our data show that the pretherapeutic optic nerve impairment revealed by the visual acuity testing is the most important factor to be considered for the functional outcome prediction of fSRT for ONSM. Due to its very low morbidity profile, fSRT should therefore be adopted early after diagnosis of ONSM.

## Figures and Tables

**Table 1 cancers-13-00522-t001:** Mean pretreatment epidemiological and clinical variables with their standard deviation (STDV) or the percentage of the variable, overall and dichotomized (favorable outcome with improvement of the acuity vs. unfavorable outcome with stable or worsening of the acuity) *p* value of the Mann–Whitney test or Fisher’s exact test between the favorable and unfavorable group.

Variable	Overall (*n*:13)	Favorable (*n*:8)	Unfavorable (*n*:5)	*p* Value
age [years, mean (STDV)]	45.3 (13.5)	44.2 (11.6)	47.1(17.5)	0.8
male gender [(%)]	53	43	60	0.9
tumor volume [ccm, mean (STDV)]	1.97 (0.81)	2.01 (0.91)	1.91 (0.76)	1
posterior tumor localisation [(%)]	62	57	80	0.9
pretreatment visual acuity [mean (STDV)]	0.38 (0.26)	0.52 (0.2)	0.14 (0.17)	0.03
total dose [Gy, mean (STDV)]	47.8 (10.2)	50.7 (0.49)	43.1 (16.2)	0.15
fractions [n, mean (STDV)]	26 (8)	28 (1)	23 (12)	0.7
retina dose [Gy, mean (STDV)]	39.3 (14.6)	39.6 (14.7)	38.9 (16.1)	1
follow up [years, mean (STDV)]	2.7 (1.5)	2.4 (1.1)	3.2 (2)	0.4

**Table 2 cancers-13-00522-t002:** Pretreatment epidemiological and clinical variable of the data review, overall and dichotomized (favorable outcome with improvement of the acuity vs. unfavorable outcome with stable or worsening of the acuity), p value of the Mann–Whitney test or Fisher’s exact test between the favorable and unfavorable group.

Variable	Overall (*n*:102)	Favorable (*n*:69)	Unfavorable (*n*:33)	*p* Value
age [years, mean (STDV)]	49.8 (11.5)	49.4 (11.5)	50.5 (11.8)	0.6
male gender [(%)]	14	10	21	0.14
tumor volume [ccm, mean (STDV)]	1.9 (0.8)	1.85 (0.93)	1.91 (0.76)	0.8
posterior tumor localisation [(%)]	59	53	65	0.7
pretreatment visual acuity [mean (STDV)]	0.45 (0.35)	0.49 (0.34)	0.36 (0.36)	0.06
total dose [Gy, mean (STDV)]	48.1 (11.4)	48,4 (10.8)	47.2 (12.8)	0.14
fractions [*n*, mean (STDV)]	25 (9)	25 (9)	25 (8)	0.6
retina dose [Gy, mean (STDV)]	26.2 (21.3)	19.5 (19.5)	43.6 (16)	0.02
follow up [years, mean (STDV)]	6.2 (4.8)	6.0 (4.5)	6.6 (5.4)	0.8

**Table 3 cancers-13-00522-t003:** Multivariate analysis of pretreatment epidemiological and clinical findings regarding favorable visual outcome.

Variable	Partial Coeficient (B)	Standard Error	*p* Value
age [years]	-	-	-
male vs. female gender	-	-	-
tumor volume [ccm]	-	-	-
anterior vs. posterior	-	-	-
pretreatment visual acuity	+0.59	0.16	<0.01
total dose [Gy]	-	-	-
fractions [n]	-	-	-
retina dose [Gy]	−0.01	0.003	<0.01
follow up [years]	−0.03	0.012	<0.05

Abbreviations: fSRT = fractionated stereotactic radiotherapy; MRI = magnetic resonance imaging; ONSM = optic nerve sheath meningioma; SRS = radiosurgery; FU = follow up; GTV = gross tumor volume, CTC = common terminology criteria; Gy = Gray.

## Data Availability

The meta-analysis data are publicly available from the primary publications. Our primary data are inserted in the article. The processed data evaluated and presented in this study are available on request from the corresponding author.

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
