# Peer review of "Prognostic Factors for Functional Outcome of Patients with Optic Nerve Sheath Meningiomas Treated with Stereotactic Radiotherapy–Evaluation of Own and Meta-Analysis of Published Data"

_cancers, 2021, doi:10.3390/cancers13030522_

Round 1

Reviewer 1 Report

Authors investigated prognostic factors for a favorable outcome (improvement of the visual acuity or visual fields) after fractionated stereotactic radiotherapy (fSRT) of optic nerve sheath meningioma (ONSM). They reported that FSRT for ONSM achieves in over 50% of cases an improvement of the ophthalmic status with low morbidity and excellent tumor control in their series and the meta analysis. Patients with a favorable outcome had in all analysis a significantly higher visual acuity before treatment start. Therefore, we advocate using fSRT as early as possible before vision deterioration occurs.

This paper is interesting. However following points should be shown.

  1. Authors should describe the natural history of ONSM.
  2. Since widespread opinion is that observation may be the best management, authors should describe that a functional improvement in the patient under observation
    can be achieved up to a cut off acuity of 0.4 in abstract. 

Author Response

Dear C

Dear Colleagues,

Thank you for your valuable correction. We tried to fulfill all the requests point by point as listed below and believe that the quality of the publication grew. We kindly ask for a reevaluation.

With Best Regards

Bogdan Pintea

Requests:

Request 1

We address the natural history of ONSM in the introduction

“It has been shown though that without any treatment over years visual impairment occurs in the vast majority of cases accompanied with a substantial risk of involvement of both eyes [ 7, 17, 18, 32], though ONSM cause no deaths.” Page 2

Request 2

We stress the consequences of our findings:

“Although our data indicate that a functional improvement can be achieved up to a cut off visual acuity of 0.4, and ROC of the data review analyses reveal a possible threshold at a visual acuity of about 0,28, the authors argue for the earliest possible start of fSRT after individual consideration of the benefits and risks.” Page 7

Reviewer 2 Report

The paper affords a very important question in neuro-oncology: the treatment ofoptic nerve sheath meningiomas. To evaluate prognostic factors for a favorable outcome after fractionated stereotactic radiotherapy the Authors performed both a clinican trial on 13 patients and a meta-analyis of published data between 2010 and 2019. However some points need to be clarified.

  1. a) in Table 1 several epidemiological and clinical parameters have been compared between favourable and unfavorable groups. Authors wrote “We found only one significant difference (p=0.02) between favorable and less favorable cases, when the visual acuity before SRT/SRS was taken into account.” Why P value was not corrected according to correction procedures such as Bonferroni’s rule for multiple comparisons ? Why T student ,a parametric test instead of Mann Whitney a non parametric test ,was chosen ?
  2. b) The most important issue of the paper is that “patients (according both to clinical trial and meta-analysis) with a favorable outcome had a significantly higher visual acuity before treatment start. It is possible, using ROC (Receiver Operating Characteristic) to set a thereshold ? If not why ?
  3. d) The meta-analyis showed that shorter FU time was significantly linked to favourable oucome. What does it mean ? Is it possible that radiogenic optic neuropathy, a severe possible side effect of radiotherapy, occurred during the follow-up ?

This important issue should be discussed. Furthermore, on my opinion, data on side effects reported in clinical trial and meta-analysis should be added.

Author Response

Dear Colleagues,

Thank you for your valuable correction. We tried to fulfill all the requests point by point as listed below and believe that the quality of the publication grew. We kindly ask for a reevaluation.

With Best Regards

Bogdan Pintea

Requests:

Request 1

We performed a multivariate analysis which is a alternative model to deal with multiple testing than the Bonferroni correction.

Thank you for your suggestion to use the Mann Whitey test, a non parametric the Mann Whitey test test  for the univariate analyses.  We performed the Mann Whitey test and for “own data” and the metaanalysis of the published data, however no major changes of the outcome could be detected. 

The Tables 1 and 2 were adjusted.

Request  2

We perfomed as requested a ROC analyse. A possible threshold might be at the visual acuity of 0,28. (Supplement 2)

We adjust the text as followed:

“MedCalc v19.0.4 software was used for Receiver operating characteristic ROC analyses.” Page 4

 “A ROC analysis reveals that the optimal threshold might be a pretreatment visual acuity of 0.28, taking in account that higher specificity is desired in our model (Supplement 2).” Page 5

“Although our data indicate that a functional improvement can be achieved up to a cut off visual acuity of 0.4, and ROC of the data review analyses reveal a possible threshold at a visual acuity of about 0,28, the authors argue for the earliest possible start of fSRT after individual consideration of the benefits and risks.” Page 7

Request  3

We address this topic:

“A shorter FU time significantly correlates with a better outcome in the multivariate analysis. The correlation is weak, however it should be carefully monitored in following studies as it might reflect a possible radiogenic optic neuropathy or a local tumor pro-gress.” Page 7

Round 2

Reviewer 2 Report

The manuscript has improved; the Authors answered to all my questions.